# Local Sustained GM-CSF Delivery by Genetically Engineered Encapsulated Cells Enhanced Both Cellular and Humoral SARS-CoV-2 Spike-Specific Immune Response in an Experimental Murine Spike DNA Vaccination Model

**DOI:** 10.3390/vaccines9050484

**Published:** 2021-05-10

**Authors:** Rémi Vernet, Emily Charrier, Erika Cosset, Sabine Fièvre, Ugo Tomasello, Julien Grogg, Nicolas Mach

**Affiliations:** 1Department of Oncology, Geneva University Hospitals and Medical School, 1211 Geneva, Switzerland; emily.charrier@unige.ch (E.C.); nicolas.mach@hcuge.ch (N.M.); 2Center for Translational Research in Onco-Hematology, Division of Oncology, Geneva University Hospitals and University of Geneva, 1211 Geneva, Switzerland; erika.cosset@unige.ch; 3MaxiVAX SA, 1202 Geneva, Switzerland; jgrogg@maxivax.ch; 4Department of Basic Neurosciences, University of Geneva, 1211 Geneva, Switzerland; sabine.fievre@unige.ch (S.F.); ugo.tomasello@unige.ch (U.T.)

**Keywords:** SARS-CoV-2, COVID 19, DNA vaccine, GM-CSF, adjuvant

## Abstract

Severe acute respiratory syndrome coronavirus 2 (SARS-CoV-2) has caused a worldwide pandemic with recurrences. Therefore, finding a vaccine for this virus became a priority for the scientific community. The SARS-CoV-2 spike protein has been described as the keystone for viral entry into cells and effective immune protection against SARS-CoV-2 is elicited by this protein. Consequently, many commercialized vaccines focus on the spike protein and require the use of an optimal adjuvant during vaccination. Granulocyte-macrophage colony-stimulating factor (GM-CSF) has demonstrated a powerful enhancement of acquired immunity against many pathogens when delivered in a sustained and local manner. In this context, we developed an encapsulated cell-based technology consisting of a biocompatible, semipermeable capsule for secretion of GM-CSF. In this study, we investigated whether murine GM-CSF (muGM-CSF) represents a suitable adjuvant for SARS-CoV-2 immunization, and which delivery strategy for muGM-CSF could be most beneficial. To test this, different groups of mice were immunized with intra-dermal (i.d.) electroporated spike DNA in the absence or presence of recombinant or secreted muGM-CSF. Results demonstrated that adjuvanting a spike DNA vaccine with secreted muGM-CSF resulted in enhancement of specific cellular and humoral immune responses against SARS-CoV-2. Our data also highlighted the importance of delivery strategies to the induction of cellular and humoral-mediated responses.

## 1. Introduction

The severe acute respiratory syndrome coronavirus 2 (SARS-CoV-2) genome encodes four structural proteins: the spike protein (S), the membrane glycoprotein (M), the nucleocapsid protein (N), and the envelope protein (E). The viral spike protein binds its receptor, angiotensin-converting enzyme 2 (ACE2), which leads to the recognition and entry of the virus into targeted cells. The spike protein S consists of two subunits: S1 contains a receptor-binding domain (RBD), which interacts with the ACE2 receptor on the cell surface, while S2 mediates the fusion of viral and cell membranes [1,2]. A study of SARS-CoV in hamsters concluded that effective immunity against SARS-CoV infection is only elicited by the S protein [3]. Most of the vaccination strategies, relying on a single protein, focus on the S protein.

To protect against SARS-CoV-2 infection, it is important to elicit neutralizing antibodies targeting the S1 RBD, S1 N-terminal domain, or the S2 region. These antibodies must block the binding of the RBD to the ACE2 receptor and prevent S2-mediated membrane fusion or entry into the host cell, thereby inhibiting viral infection [4,5]. Previous studies of vaccines for MERS-CoV (Middle East Respiratory Syndrome Coronavirus) [6] and SARS-CoV [7] showed that both humoral and cellular (cytotoxic) immune responses are important to inducing a protective immune response. To achieve these outcomes, one possible option is deoxyribonucleic acid (DNA) vaccination.

Among the various vaccination approaches against infectious diseases, such as human immunodeficiency virus (HIV), DNA vaccines have several advantages: they are easily produced, they provide opportunities for molecular engineering, they lack anti-vector immunity, and they have the potential to promote both cellular and humoral immune responses [8,9,10]. However, despite their high immunogenicity in murine models, DNA vaccines have shown poor efficiency in large animal models and in humans [11]. Interestingly, a study showed that a DNA vaccine encoding the S protein of SARS-CoV induced T cell and neutralizing antibody responses, which protected mice against an intranasal SARS-CoV challenge [12]. New strategies for the improvement of DNA vaccines include: optimization of transcriptional control elements; use of adjuvants such as toll-like receptor (TLR) ligands [13,14], cytokine-expressing plasmids [15,16,17], or apoptosis-based adjuvants [18]; and appropriate delivery systems, such as local electroporation (EP) [19,20,21].

Granulocyte-macrophage colony-stimulating factor (GM-CSF) has demonstrated significant adjuvant effect when included in DNA vaccines for many infectious diseases [22,23,24,25,26,27,28,29]. For example, the co-administration of plasmid-encoded GM-CSF with an HIV-1 DNA vaccine improved both the magnitude and quality of vaccine-induced T-cell responses, particularly by increasing proliferating CD4+ T cells which simultaneously produce interferon-γ, tumor necrosis factor-α, and interleukin-2 [30]. In a murine model of the Ebola virus, the inclusion of interleukin-12 or GM-CSF improved cell-mediated immunity. However, the GM-CSF adjuvant plasmid did not improve neutralizing antibody titers in this model [26]. The combination of GM-CSF with self-amplifying mRNA constructs encoding the influenza virus nucleoprotein significantly improved the magnitude of antigen-specific CD8+ T cell responses and increased recruitment of antigen-presenting cells at the vaccination site [27]. These results imply a local release of GM-CSF. Indeed, the delivery system of GM-CSF seems to have an impact on the initiation of immune response. Reali et al. [31] observed an increase in the immune response in mice following the local administration of GM-CSF. In contrast, systemic administration of GM-CSF results in immune suppression through the production of myeloid-derived suppressor cells (MDSCs) [32].

To allow sustained local delivery of GM-CSF as an adjuvant in an anti-cancer vaccine, we developed an encapsulated cell technology in which genetically modified cells are loaded into a biocompatible, semipermeable capsule which can secrete human or murine GM-CSF. This medical device is co-implanted in the subcutaneous tissue at the site of antigen injection. This delivery system allows GM-CSF administration which induces recruitment and early maturation of dendritic cells, a critical step for vaccine efficacy. Accordingly, we hypothesized that adjuvanting a spike DNA vaccine with muGM-CSF may result in an enhancement of anti-SARS-CoV-2 immune response. We evaluated whether local release of muGM-CSF in combination with a spike DNA vaccine leads to an increase in both cellular and humoral immune responses.

## 2. Materials and Methods

### 2.1. Animals

Ten-week-old C57BL/6J female mice were purchased from Charles River Laboratories (Saint-Germain-sur-l’Arbresle, France). Animals were group housed in a specific pathogen-free environment, at 21 °C and 29% humidity, with lighting on a 12 h light/dark cycle. Mice were allowed to acclimate for at least 2 weeks before any manipulation.

### 2.2. Plasmid Amplification, Purification, and Validation

The SARS-CoV-2 spike protein-coding plasmid was kindly provided by Professor Gary Kobinger (University of Laval, Québec City, QC, Canada).

Plasmid DNA was amplified using Subcloning Efficiency DH5α Competent Cells (Thermo Fisher Scientific, Waltham, MA, USA, 18265017) grown in LB medium containing 100 μg/mL ampicillin (AppliChem, Darmstadt, Germany, A0839) and 100 μg/mL kanamycin (AppliChem A1493). Plasmid DNA was then purified using an EndoFree Plasmid Mega Kit (Qiagen, Hilden, Germany, 12381).

The amplified plasmid was finally analyzed by EcoRI digestion followed by gel electrophoresis, and was also fully sequenced using the iGE3 Genomics Platform of the University of Geneva using Illumina HiSeq 4000 technology.

### 2.3. Plasmid Injection and Electroporation

On shaved skin, each animal received intradermal injections of 23 μL of the spike protein-coding plasmid (2 μg/μL) diluted in sterile phosphate-buffered saline (PBS) and followed by EP. EP was performed by applying 5 electric pulses (15 V for 50 ms at 1 Hz) with a square-wave electroporator (Nepa Gene, Sonidel Limited, Dublin, Ireland).

### 2.4. Dermal SARS-CoV-2 Spike Protein Expression by Western Blot Analysis

Skin punch biopsies were collected and mechanically dissociated in gentleMACS M Tubes (Miltenyi Biotech, Bergisch Gladbach, Germany, 130-093-236) using the gentleMACS protein program. Tissue samples were processed in 500 µL of T-PER Tissue Protein Extraction Reagent (Thermo Fisher Scientific, Waltham, MA, USA, 78510) and cOmplete, Mini, EDTA-free Protease Inhibitor Cocktail Tablets (Ethylenediaminetetraacetic acid) (Sigma Aldrich, St. Louis, MO, USA, 04693159001). After dissociation, samples were centrifuged at 3300× *g* for 10 min at 4 °C, then the supernatant was collected and analyzed for protein quantification. Protein concentration was determined using a Microassay BCA Protein Assay Kit (Bicinchoninic Acid Assay) (Thermo Fisher Scientific, 23235) with bovine albumin serum used as the standard. Digested samples were loaded on a 6% SDS-PAGE gel, then transferred overnight (O/N) at 4 °C on a methanol-activated PVDF membrane followed by a blocking step with PBS + 0.05% Tween 20 (PBST) + 5% BSA for 1 h at room temperature (RT). Anti-beta-actin HRP antibody was used as a loading control. Membranes were incubated O/N at 4 °C under gentle agitation with 1:500 SARS-CoV/SARS-CoV-2 spike protein (S-RBD) Chimeric Recombinant Mouse Monoclonal Antibody (D005) (Thermo Fisher Scientific, MA5-35958) diluted in blocking buffer. After washing, membranes were incubated in a Mouse anti-Human IgG1 Fc Secondary Antibody, HRP (Thermo Fisher Scientific, A-10648) solution for 2 h at RT for detection. Extensive washing was done before revelation using SuperSignal West Pico PLUS Chemiluminescent Substrate (Thermo Fisher Scientific, 34577). Image acquisition was performed with Syngene PXi reader (Syngene, Cambridge, UK).

### 2.5. muGM-CSF Adjuvant Administration

Two distinct techniques were used for the delivery of muGM-CSF: administration of recombinant muGM-CSF, or, delivery via an implanted clinical grade medical device (or “capsule”) containing muGM-CSF-producing cells. For recombinant cytokine administration, 4.2 µg of muGM-CSF recombinant protein (Peprotech, Rocky Hill, NJ, USA, 315-03) was injected subcutaneously with a 30G syringe at the immunization site. For sustained delivery, biocompatible capsules were each loaded with 106 cells genetically engineered to secrete muGM-CSF via a loading tube (BD Vasculon, BD Biosciences, San Jose, CA, USA). The loading tube was cut and capsules were sealed using Ultra-violet -sensitive adhesive glue (Dymax, Wiesbaden, Germany). Capsules were then placed in a 12-well plate in 2 mL of growth medium for 24 h prior to implantation.

For the implantation, mice were anesthetized with isoflurane and a small incision was performed on the lower back of the animal. Capsules were implanted subcutaneously in the flank of the animals, through a 14G catheter. The incision was closed with surgical staples and animals recovered in their home cage. Analgesia was provided by subcutaneous injection of buprenorphine (0.1 mg/kg) prior to implantation and paracetamol (2 mg/mL) in drinking water for 3 days. One week after the implantation, animals were anesthetized using isoflurane and the medical device was removed via a small incision.

### 2.6. muGM-CSF Adjuvant Quantification by ELISA

The secreted muGM-CSF adjuvant was quantified using a GM-CSF Murine ELISA (Enzyme-Linked Immunosorbent Assay) Kit (Thermo Fisher Scientific, BMS612) according to the manufacturer’s recommendations.

### 2.7. Immunization Scheme

At Day 0, mice were immunized with the DNA plasmid coding for SARS-CoV-2 spike protein (Figure 1A) with or without muGM-CSF as an adjuvant. There were 3 treatment groups of 12 mice each. Mice in the first group received injection of the DNA plasmid without any adjuvant. Mice in the second and third groups were also immunized with the DNA plasmid adjuvanted with either recombinant muGM-CSF or muGM-CSF secreted by the encapsulated cell line, respectively. On the day of immunization, a submandibular blood puncture was performed to isolate serum at baseline. On Days 10 and 28, 5 mice per group were sacrificed. The 2 remaining mice per group received an additional immunization with plasmid DNA and were kept for a further 28 days until sacrifice. At sacrifice, a skin punch biopsy was taken from all animals, digested, and analyzed by western blot for the presence of the SARS-CoV-2 spike protein.

### 2.8. T Cell Characterization by Intracellular Cytokine Staining (ICS) and Flow Cytometry Analysis

Spleens were harvested, homogenized, passed through a 70-µm cell strainer (Falcon, Corning, NY, USA), and incubated with ACK Lysing Buffer (Thermo Fisher Scientific, A1049201). Cells were washed and resuspended in RPMI 1640 medium (Thermo Fisher Scientific, 72400021) containing 10% heat-inactivated FBS (Thermo Fisher Scientific, 10101-145) and 1% penicillin-streptomycin (Thermo Fisher Scientific, 15140122). Splenocytes were added to 5 mL tubes at a concentration of 10^6^ cells/mL and stimulated for 12 h in the presence of BD Golgi Plug Protein Transport Inhibitor (BD Bioscience, 51–2301 KZ) for the last 9 h under a controlled atmosphere (5% CO_2_ and 37 °C), with 1 µg/10^6^ cells of SARS-CoV-2 spike-specific peptides. Overlapping peptides (15-mers with an 11-amino-acid overlap) were synthetized from the spike glycoprotein sequence by the Peptides Core Facility, University of Geneva. The 316 peptides generated were randomly distributed in 2 pools (Pool 1 and Pool 2) (Appendix A). DMSO-stimulated cells served as a background control and cells stimulated with 50 ng/mL PMA (Sigma Aldrich, P8139) and 750 ng/mL ionomycin (Sigma Aldrich, C9275) were used as positive controls.

After stimulation, cells were washed with PBS, blocked with CD16/CD32 (BD Biosciences, San Jose, CA, USA, 553142), and cell surface antigens were stained using PE-Cy7-conjugated anti-mouse CD107a (clone: 1D4B, Thermo Fisher Scientific, 25-1071-82), BV510-conjugated anti-mouse NK1.1 (clone: PK136, BD Bioscience, 563096), SB645-conjugated anti-mouse CD4 (clone: RM4-5, Thermo Fisher Scientific, 64-0042-82), SB780-conjugated anti-mouse CD8a (clone: 53-6.7, Thermo Fisher Scientific, 78-0081-82), PE-CF594-conjugated anti-mouse Ly6g/Ly6c (clone: RB6-8C5, BD Bioscience, 562710), PE-CF594-conjugated anti-mouse CD45R (clone: RA3-6B2, BD Bioscience, 562290), and BV510-conjugated anti-mouse CD49b (clone: HMα2, BD Bioscience, 740133) antibodies for 30 min at 4 °C in the dark. After incubation, cells were washed then fixed and permeabilized using a BD Cytofix/Cytoperm kit (BD Bioscience, 554714). After blocking with CD16/CD32, cells were finally stained for intracellular antigens with BV421-conjugated anti-mouse CD3 (clone: 17A2, BD Bioscience, 564008), PerCP-CY 5.5-conjugated anti-mouse TNFα (clone: MP6-XT22, BD Bioscience, 560659), APC-conjugated anti-mouse IL2 (clone: JES6-5H4, Thermo Fisher Scientific, 17-7021-81), FITC-conjugated anti-mouse IFNγ (clone: XMG1.2, BD Bioscience, 554411), and PE-conjugated anti-mouse IL10 (clone: JES5-16E3, Thermo Fisher Scientific, 12-7101-82) antibodies. All staining steps were performed using BD Horizon Brilliant Stain Buffer (BD Bioscience, 566349). Following final washes, cells were resuspended in PBS and analyzed using a BD LSRFortessa Flow Cytometer (BD Bioscience). The percentages of cytokine-secreting CD4+ or CD8+ T cells were obtained using Facs DIVA software and, for the polyfunctionality analysis, the frequencies were corrected by background subtraction (DMSO negative condition). A minimum of 20,000 CD4+ T cells and 5000 CD8+ T cells were analyzed.

### 2.9. Quantification of SARS-CoV-2 Spike Protein Antibodies by ELISA

Clear Flat-Bottom Immuno Nonsterile MEDISORP 96-Well Plates (Thermo Fisher Scientific, 467320) were coated with 100 ng of SARS-CoV-2 spike protein S1/S2 (S-ECD) (aa11-1208) (Thermo Fisher Scientific, RP87680) in 100 μL of PBS per well, O/N at 4 °C. Plates were washed 4 times for 5 min with wash buffer (Thermo Fisher Scientific, 3501004) and then blocked with PBST + 1% BSA for 1 h at RT under gentle agitation. Serum samples were diluted 1:100 in 100 μL blocking buffer and incubated for 90 min at RT under gentle agitation. After washing, Goat anti-Mouse IgG, IgM, IgA (H + L) Secondary Antibody, HRP (Thermo Fisher Scientific, A-10668) or Goat anti-Mouse IgG (H + L) Cross-Adsorbed Secondary Antibody, HRP (Thermo Fisher Scientific, G-21040) were used for the detection of bound antibodies and plates were incubated for 1 h at RT under gentle agitation. Extensive washing was done before chromogenic revelation using TMB substrate (Thermo Fisher Scientific, SB01) for 30 min at RT in the dark and stopped with a stop solution (Thermo Fisher Scientific, SS01). Absorbance of the samples was immediately measured at 450 nm. During each analysis, a standard curve was produced using S-RBD Chimeric Recombinant Mouse Monoclonal Antibody (D005) (Thermo Fisher Scientific, MA5-35958) and Mouse anti-Human IgG1 Fc Secondary Antibody, HRP (Thermo Fisher Scientific, A-10648). The range of this standard curve was between 1 μg/mL and 1.953 ng/mL.

### 2.10. SARS-CoV-2 Surrogate Virus Neutralization Test (sVNT)

A neutralizing assay was performed using the SARS-CoV-2 Surrogate Virus Neutralization Test (sVNT) (Genscript, U7290FJ010). The positive control, negative control, and serum samples were mixed in independent vials with the HRP-RBD solution at a final ratio of 1:1. The samples were incubated at 37 °C for 30 min and then 100 µL of each preparation was loaded on the 96-well plate and incubated at 37 °C for 15 min. After incubation, plates were washed 4 times with 1× wash solution. Chromogenic revelation was done using 100 µL of TMB substrate (Thermo Fisher Scientific, SB01) which was added to each well before incubation at RT in the dark for 15 min. The reaction was quenched with 50 µL of stop solution (Thermo Fisher Scientific, SS01) per well. Absorbance of the samples was measured immediately at 450 nm.

### 2.11. Statistical Analysis

Data analyses were performed using GraphPad Prism (version 8.1.1., GraphPad Software). For cytokine quantification, a two-way ANOVA (Analysis of variance) with Tukey’s test was used for analysis of multiple groups. For ELISA analysis, a one-way ANOVA was used with: Bartlett’s test for samples at Day 10 and 28; the Brown-Forsythe test at Day 56 for serum analysis; and a paired t test for the muGM-CSF secretion by encapsulated cells. Differences regarded as statistically significant between the groups are presented as follows: * *p* < 0.05, ** *p* < 0.01, *** *p* < 0.001, and **** *p* < 0.0001.

## 3. Results

### 3.1. Validation of the Vaccination Strategy

To investigate the role of muGM-CSF as an adjuvant in SARS-CoV-2 immunization and the effect of delivery method, our vaccination strategy was based on an injection of a DNA plasmid coding for SARS-CoV-2 spike protein as a source of antigen, with or without muGM-CSF as an adjuvant (Figure 1A). Mice in the first group received injection of the DNA plasmid without any adjuvant. Mice in the second and third groups were also immunized with the DNA plasmid adjuvanted with muGM-CSF (recombinant or secreted by encapsulated cells).

To assess DNA electroporation efficiency, western blot analysis was performed on skin punch lysate for each animal included in the study. This analysis revealed the presence of SARS-CoV-2 spike protein in the skin of all animals at Days 10 and 56, and almost all of the animals at Day 28 (Figure 1B and Appendix A shows the uncropped western blot). Of note, we observed variations between samples due to the reliability of both localizing the injection site after several days and homogenizing the samples. Therefore, levels of SARS-CoV-2 spike protein could not be evaluated quantitatively.

As a validation step, we used ELISA to analyze the muGM-CSF secreted by our medical device before implantation and after explantation. Encapsulated cells secreted 929.5 ± 67.7 ng muGM-CSF/24 h/capsule (mean ± Standard Deviation) prior to implantation. After 7 days in vivo and following explantation, the average secretion from the capsules was 25.5 ± 19.9 ng muGM-CSF/24 h/capsule. Importantly, this decrease in muGM-CSF production is expected given the exhaustion of the encapsulated cells after 7 days due to the progressive recruitment of inflammatory cells around the medical device.

### 3.2. Cellular-Mediated Immune Responses to SARS-CoV-2 Spike Protein

To investigate whether our strategy induces a cellular immune response against the spike protein, mice were immunized with a single dose of spike DNA plasmid with or without muGM-CSF (recombinant or secreted by encapsulated cells) as an adjuvant. T cell response was measured at Day 10 by intracellular cytokine staining (ICS) after stimulation of splenocytes with spike protein peptide pools (Appendix A shows the gating strategy).

We observed a significant increase in the percentage of TNF-α- and IFN-γ-secreting CD8+ T cells after spike peptide Pool 1 stimulation for the three vaccination strategies (DMSO without adjuvant: 0.018 ± 0.007; Pool 1 without adjuvant: 0.078 ± 0.014; DMSO with recombinant muGM-CSF: 0.037 ± 0.010; Pool 1 with recombinant muGM-CSF: 0.115 ± 0.027; DMSO with capsule muGM-CSF: 0.054 ± 0.029; and Pool 1 with capsule muGM-CSF: 0.113 ± 0.023). Since the background (DMSO condition) was slightly increased when muGM-CSF was added to the vaccine (recombinant muGM-CSF: 0.037 ± 0.010 and capsule muGM-CSF: 0.054 ± 0.029) compared with the non-adjuvanted group (without adjuvant: 0.018 ± 0.007), no significant difference was observed between the three groups in response to spike peptide Pool 1 (Figure 2).

Spike peptide Pool 2 induced significant cytokine secretion only when the DNA vaccine was adjuvanted with muGM-CSF. Indeed, in the group implanted with capsules, we observed a slight increase in the percentage of TNF-α-secreting CD8+ T cells after spike peptide Pool 2 stimulation (DMSO condition: 0.047 ± 0.024 and Pool 2: 0.153 ± 0.062). Moreover, adjuvanted immunization with recombinant muGM-CSF did not quantitatively amplify the cellular response compared with the non-adjuvanted group (Figure 2).

To further investigate whether the addition of muGM-CSF can modify the secretion profile of CD8+ T cells, the polyfunctionality of CD8+ T cells was analyzed. The results showed proportion of functional CD8+ T cells after stimulation (Figure 3). Moreover, we observed an increase in the polyfunctionality of the CD8+ T cells when they were isolated from animals immunized with an adjuvanted vaccine (Figure 3 and Appendix A).

No cytokine secretion after spike peptide pool stimulation was observed for the CD4+ T cells (Figure 4). muGM-CSF secreted by encapsulated cells increased TNF-α and IL-2 production even in the negative condition, suggesting a non-specific reactivity of CD4+ T cells.

### 3.3. Humoral-Mediated Immune Responses to SARS-CoV-2 Spike Protein

Humoral responses were assessed by ELISA quantification of circulating antibodies present in the animal’s blood. Two distinct analyses were performed on serum samples: quantification of the total amount of antibodies (IgG, IgM, IgA) directed against the SARS-CoV-2 spike protein, and quantification of the amount of IgG directed against the SARS-CoV-2 spike protein. Baseline pre-immunization serum values were used as a specific background per animal and a fold-change value was calculated between baseline samples and samples harvested on the day of sacrifice. Regarding the total amount of antibodies (IgG + IgM + IgA), results showed an effect of the muGM-CSF-adjuvanted vaccination after 28 days compared with the non-adjuvanted group (Figure 5A). All mice from the muGM-CSF-adjuvanted groups developed antibodies against the SARS-CoV-2 spike protein after 28 days. In contrast, in the non-adjuvanted group, only 2 out of 5 developed antibodies after 28 days. It is likely this difference was driven by IgM and/or IgA, as there was no difference when only IgG was analyzed (Figure 5B). At Day 56, we observed a significant increase in the amount of both total antibodies and IgG in mice immunized with spike DNA and capsules containing muGM-CSF-producing cells.

Another interesting observation came from these data when comparing the two adjuvant methods. As previously described in the literature, we observed a deleterious effect of muGM-CSF delivered as recombinant protein at a high dose [32], in contrast with the immunostimulatory effect of the low, sustained, and local delivery by encapsulated cells [33,34]. Indeed, the fold-change of IgG, IgM, and IgA in the DNA spike + recombinant muGM-CSF group was lower than the group without adjuvant. The same pattern was observed for IgG alone.

Finally, to determine if these generated antibodies have a potential protective effect, serum samples were tested for their neutralization potency. Unfortunately, the available sVNT kit was not compatible with our murine samples, giving rise to a high background signal in the negative control group, making these data uninterpretable (data not shown).

## 4. Discussion

The use of adjuvants for efficient immunization is critical for effective therapeutic cancer vaccines, and may also play a role in some prophylactic immunization schemes against infectious diseases [35,36]. This may be of particular interest for DNA/RNA vaccines. Here, we evaluated the immunostimulatory effect of two muGM-CSF delivery methods alongside a SARS-CoV-2 spike DNA vaccine using a murine model.

Before addressing the effect of the adjuvant, we have to comment on the limited immunization capacity of the vaccine model. Indeed, i.d. immunization with electroporation of spike DNA plasmid in this murine model is suboptimal. Several aspects can explain the poor efficacy of this model. With very limited data available in the literature, the mouse is certainly not the most suitable animal model for SARS-CoV-2 immunization [37]. Firstly, SARS-CoV-2 does not use mouse ACE2 as its receptor [38] and wild-type mice are thought to be less susceptible to SARS-CoV-2 infection, although transgenic mice expressing human ACE2 have now been developed [39,40]. Secondly, the DNA plasmid in this study was administered by i.d. injection followed by electroporation. This method is not the most common delivery strategy as intramuscular injection is typically preferred, but was selected to allow proximity with the adjuvant source [12]. Finally, no optimization step for the plasmid sequence was performed and a plasmid coding for the full-length spike protein was used.

Regarding GM-CSF, it is a well-known inflammatory cytokine often used as an adjuvant in therapeutic or prophylactic cancer vaccinations. For obvious practical reasons, GM-CSF was administered as a recombinant protein, leading to the delivery of high doses and systemic activity. However, GM-CSF’s immunostimulatory effect requires specific conditions. In fact, high dose and/or systemic activity has been associated with immune tolerance via the recruitment of suppressive cells like MDSCs [32]. This effect may explain many negative trials performed with GM-CSF given in a non-stimulatory way and can be correlated with the deleterious effect observed in this study when GM-CSF was subcutaneously administrated as a recombinant protein.

In contrast, low-dose, sustained, local delivery at the vaccine site has been associated with a strong adjuvant effect of GM-CSF in both infectious disease and cancer models [31]. Strong enhancement of specific anti-infectious agents has been documented in vaccines containing a GM-CSF cassette for TB, malaria, Ebola, rabies, HPV, EBV, and HIV, among others [22,23,24,25,26,27,28,29]. Regarding cancer immunotherapy, such plasmid approaches do not yet apply as no potent tumor-specific antigens have been successfully developed at present.

To reproduce the local, low-dose, sustained delivery of GM-CSF at the vaccine site, we engineered a clinical grade cell encapsulation technology by loading cells engineered to produce muGM-CSF into a semipermeable capsule. To further document the adjuvant effect of this immuno-stimulatory cell encapsulation strategy, we evaluated this effect in a murine SARS-CoV-2 spike DNA vaccine model. The results showed that a better specific immune response, both humoral and cellular, could be obtained when muGM-CSF was delivered via the capsule compared with the non-adjuvanted and the recombinant muGM-CSF-adjuvanted groups. Interestingly, we noticed a higher cellular response when splenocytes were stimulated with SARS-CoV-2 spike-specific overlapping peptides Pool 1 compared with Pool 2; suggesting that Pool 1 contained more immunodominant SARS-CoV-2 spike peptides or immunogenic peptides recognized by murine T cells than Pool 2 did. Although the mouse may not be the best model for SARS-CoV-2 immunization, these data demonstrate that the delivery method of muGM-CSF as an adjuvant is an important factor and that cell encapsulation strategy may be of interest in other infectious diseases for which standard immunization has proven weak or ineffective.

## Figures and Tables

**Figure 1 vaccines-09-00484-f001:**
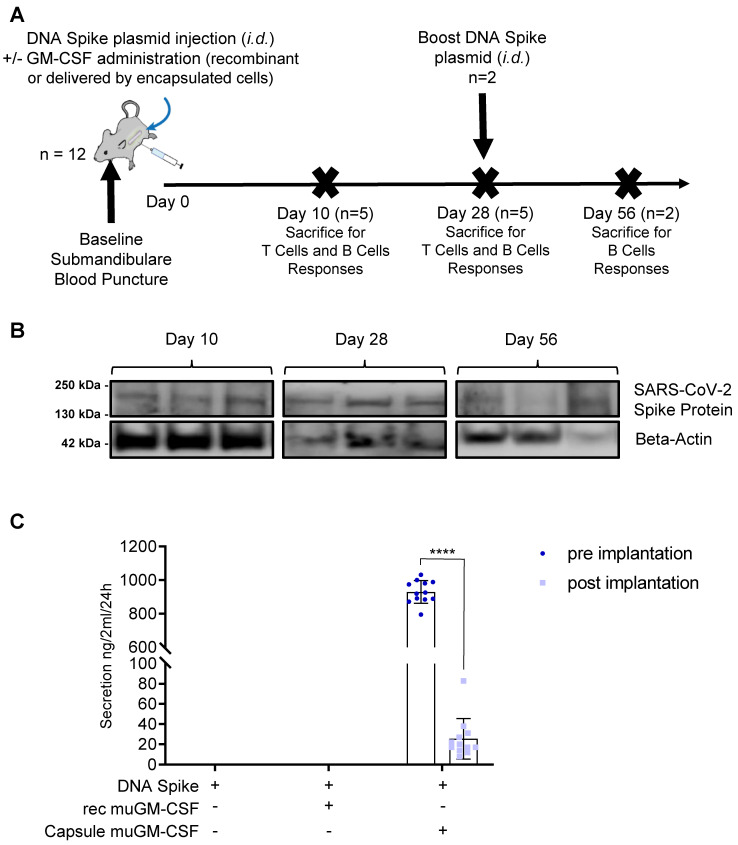
Vaccination Strategy. (**A**). Vaccination scheme. Three groups of 12 mice were immunized at Day 0 by an intradermal injection of DNA plasmid coding for SARS-CoV-2 spike protein. For two groups, the immunization was adjuvanted either with recombinant muGM-CSF or with muGM-CSF secreted by encapsulated cells. On the day of immunization, a submandibular blood puncture was performed for baseline serum isolation. On Days 10 and 28, 5 mice per group were sacrificed to assess T Cell and B Cell responses. At Day 28, the 2 remaining mice per group received a boost DNA spike plasmid injection and were kept until Day 56 for B Cell responses assessment. (**B**). Dermal SARS-CoV-2 Spike protein expression by western blot analysis. Skin punch biopsies were taken at the site of intradermal injection from all animals, digested, and analyzed by western blot for the presence of the SARS-CoV-2 spike protein. Three randomly selected mice per time points are represented. (**C**). muGM-CSF Adjuvant quantification by ELISA. The muGM-CSF secreted by encapsulated cells was quantified by ELISA before and after implantation in mice. **** *p* < 0.0001.

**Figure 2 vaccines-09-00484-f002:**
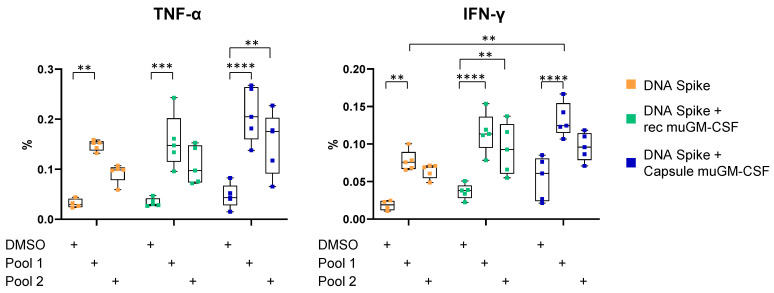
TNF-α and IFN-γ production by CD8+ T cells 10 days after vaccination. Mice were immunized with a single dose of spike DNA plasmid with or without GM-CSF (recombinant or secreted by encapsulated cells) as an adjuvant. T cells response was measured at Day 10 by intracellular cytokine staining (ICS) after stimulation of splenocytes with spike protein peptide pools or DMSO as a negative control. Data show the percentage of TNF-α or IFN-γ-secreting CD8+ T cells after stimulation. Data are represented as min-to-max boxes with individual values. ** *p* < 0.01, *** *p* < 0.001, and **** *p* < 0.0001.

**Figure 3 vaccines-09-00484-f003:**
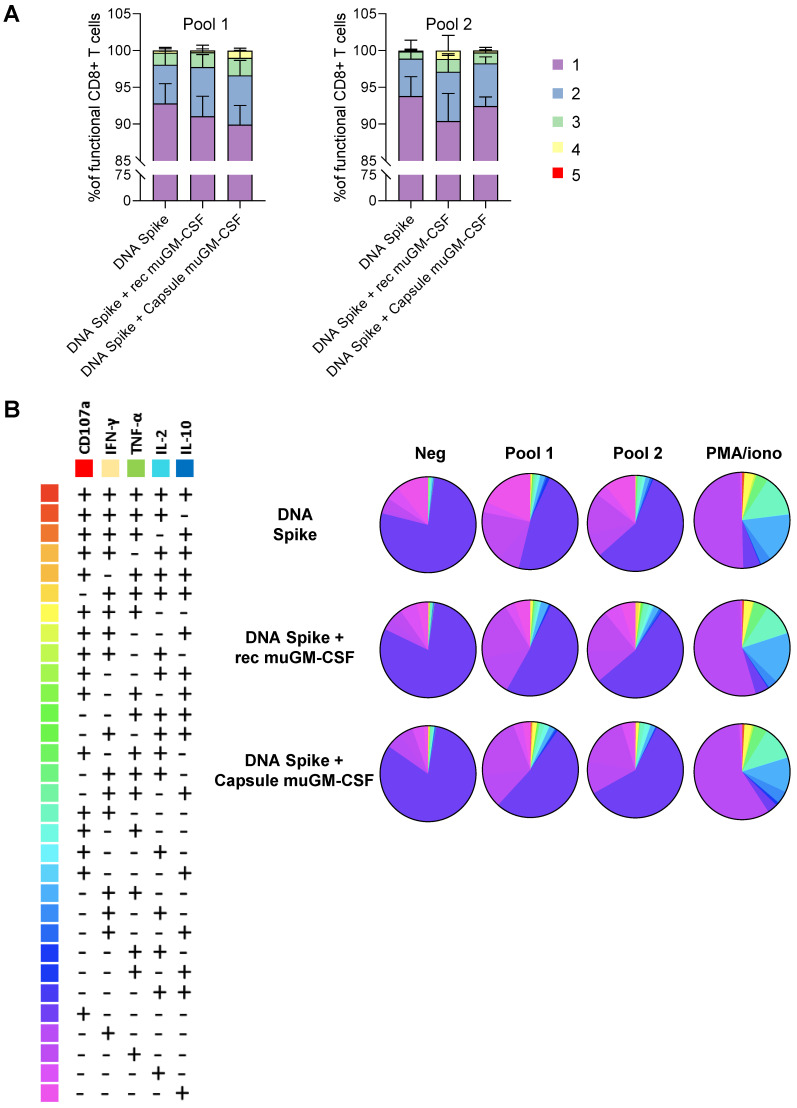
Polyfunctionality of Spike-specific CD8+ T cells at day 10 post-vaccination. (**A**). Histogram representing the percentage of functional CD8+ T cells with 1, 2, 3, 4 or 5 functions after pool 1 and pool 2 stimulation. (**B**). Pie charts representing the proportion of responding CD8+ T cells expressing different combinations of cytokines and degranulation markers after stimulation with spike peptide pool 1 or pool 2, PMA/ionomycin (positive control) or DMSO (negative control) stimulation. Histograms and Pie charts represents the mean of 5 animals per group.

**Figure 4 vaccines-09-00484-f004:**
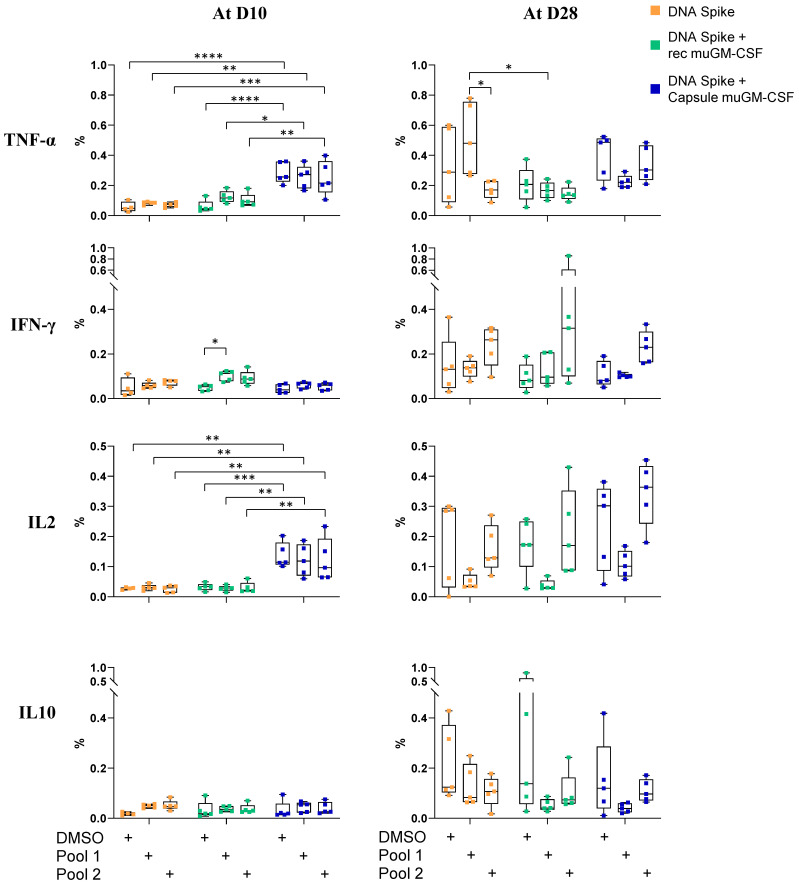
Cytokines profile of spike-specific CD4+ T cells 10 and 28 days after vaccination. Mice were immunized with a single dose of spike DNA plasmid with or without GM-CSF (recombinant or secreted by encapsulated cells) as an adjuvant. T cells responses were measured at day 10 and 28 by intracellular cytokines staining (ICS) after stimulation of splenocyte with spike protein peptide pools or DMSO as a negative control. Data show the percentage of cytokines secreting CD4+ T cells after stimulation. Data show the percentage of responding CD4+ T cells after stimulation. Data are represented as min-to-max boxes with individual values. * *p* < 0.05, ** *p* < 0.01, *** *p* < 0.001, and **** *p* < 0.0001.

**Figure 5 vaccines-09-00484-f005:**
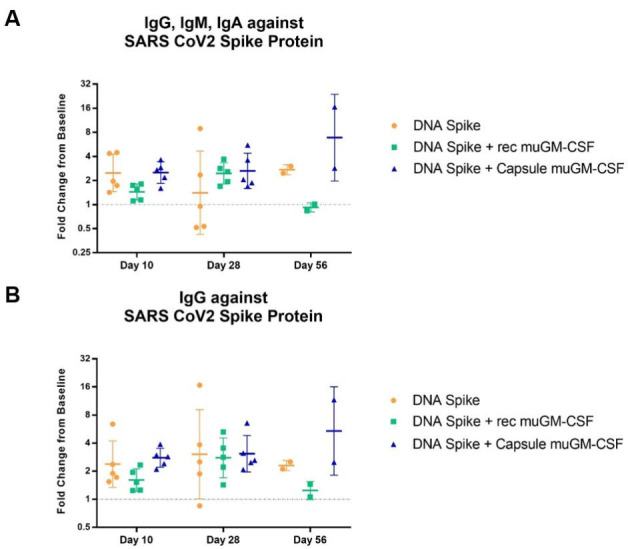
Antibodies against SARS-CoV-2 Spike Protein Detected by ELISA.Mice were immunized with a single dose of spike DNA plasmid with or without GM-CSF (recombinant or secreted by encapsulated cells) as an adjuvant. B cell response was measured at day 10 and 28 by quantification of the serum’s antibodies. (**A**). IgG, IgM, IgA against SARS CoV2 Spike Protein. Day 10, *p* value * = 0.0324; Day 28, *p* value * = 0.0277; Day 56, *p* value **** < 0.0001. (**B**). IgG against SARS CoV2 Spike Protein. Day 10, *p* value * = 0.0162; Day 28, *p* value ** = 0.0089; Day 56, *p* value **** < 0.0001. Fold change is calculated between the baseline and the day of sacrifice values. Data analysis are represented with geometric means with geometric SD.

## Data Availability

Data can be accessed upon the request.

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
