# Peer review of "Local Sustained GM-CSF Delivery by Genetically Engineered Encapsulated Cells Enhanced Both Cellular and Humoral SARS-CoV-2 Spike-Specific Immune Response in an Experimental Murine Spike DNA Vaccination Model"

_vaccines, 2021, doi:10.3390/vaccines9050484_

Round 1

Reviewer 1 Report

The article write by Vernet et al. is well-constructed. Some modifications need to be address. Have more information about SARS-CoV 2 are valuable. GM-CSF delivery formulation is very interesting in order to boost immune response. However, production of antibody and seroneutralization was not challenge in this study by a proper animal model of infection.

As mentioned by the authors in the discussion, the murine model was probably not fully adapted to SARS-cov 2 infection. Authors could be adding in the discussion section an appropriate mention of the limit of detection of kits especially in ELISA and for others (ng precisions)…

For the study design I m not convinced of the usage on days 56 with only 2 mices in this group. Those results are not mentioned in the supplementary data ? notably in the figure S2 ? Why?

Concerning the material and methods sections I found typo mistake or imprecisions.

Line 118 in g please not rpm

On the figure 1 please the panel B is blurred and no size marker are present. Please improve this figure.

On the figure 3 authors not indicate the proportion on labels for disk representation.

They not indicate on the figure pool 1 and 2 how many mice are concerned.

Please not change the % axis in each panel on the supplemental figure 2. Maybe just 2 different axes but not for each panel. We could think major differences and it’s probably not true.

Finally, authors need to improve their figures.

I have major concerns regarding the number of mice included and some not shown data, negative results are results please include them in the results or supplementary.

 I recommend major modifications.

Author Response

Dear Reviewer, 

 Thank you for your constructive comments.

Please find attached as a PDF our responses. 

Best Regards

Rémi VERNET

Reviewer 2 Report

The authors describe a method to increase, and therefore improve, the response to the vaccine against SARS- Cov2, both cell-mediated and humoral.

The paper is well written and the methods explained in great detail.

My only minor concern is about Pool 1 and Pool 2. I would suggest some more details on that strategy in Materials and Methods, and some comment in Discussion about the different results of the two pools.

Author Response

Dear Reviewer, 

 Thank you for your constructive comments.

Please find attached as PDF our responses. 

Best Regards

Rémi VERNET

Reviewer 3 Report

1) In figure 1B, can the authors please indicate the nearest size (kda) from the ladder. Can they also provide a positive and negative control for Spike to demonstrate that variability is not due to antibodies used. In figure 1C, can they perform statistical analyses between pre and post implantation. For all of Figure1, please provide more detail in the figure caption.

2) Please provide a figure caption for Supplemental Figure 1.

3) For figure 2, what statistical test was used? t-test? ANOVA? Is this really significant with such a small difference in percentage?

4) In figure 3A, please fix the typo in the word 'functional'. In figure 3B, please arrange the data in another way than a pie chart.

5) For figure supplemental 2 and figure 4, please clearly provide statistical analyses. I also appreciate the comparison in supplemental 2, so perhaps this could be an actual figure in the text.

6) Some graphs have information below the test. Please keep the descriptions consistent throughout the text and add more information for figures in the analyses section in the methods to describe.

7) Please make sure to spell the name of the virus consistently in the text 'SARS-CoV-2'.

8) Have you considered neutralization assays? You could use a pseudotyped virus.

Author Response

(The authors gave the same response as above.)

Round 2

Reviewer 1 Report

The authors have changed and improved all figures. However concerning the figure 1B authors need to be more precise about the western blot. There are 3 western blot by time? but in the figure legend they have mentioned all animals? I not understand

Some typo still need to be modify

Author Response

Dear Reviewer,

Thank you for your comment. We have been able to implement correction. 

The authors have changed and improved all figures. However concerning the figure 1B authors need to be more precise about the western blot. There are 3 western blot by time? but in the figure legend they have mentioned all animals? I not understand

You are right, we have decided to represent three mice per time points, the figure caption has been modified as followed : « B. Dermal SARS-CoV-2 Spike protein expression by western blot analysis. Skin punch biopsies were taken at the site of intradermal injection from all animals, digested, and analyzed by western blot for the presence of the SARS-CoV-2 spike protein. 3 randomly selected mice per time points are represented.

» Some typo still need to be modify

We have performed another round of review by a professionnal medical writer, and we cannot find any other typos.

Best Regards,

Rémi VERNET